# Genetically Predicted Levels of Serum Metabolites and Risk of Sarcopenia: A Mendelian Randomization Study

**DOI:** 10.3390/nu15183964

**Published:** 2023-09-13

**Authors:** Tingting Sha, Ning Wang, Jie Wei, Hongyi He, Yilun Wang, Chao Zeng, Guanghua Lei

**Affiliations:** 1Department of Orthopaedics, Xiangya Hospital, Central South University, Changsha 410008, China; tingtingsha@csu.edu.cn (T.S.); ningwang0405@csu.edu.cn (N.W.); hongyi_he@csu.edu.cn (H.H.); 2Key Laboratory of Aging-Related Bone and Joint Diseases Prevention and Treatment, Ministry of Education, Xiangya Hospital, Central South University, Changsha 410008, China; weij1988@csu.edu.cn; 3Hunan Key Laboratory of Joint Degeneration and Injury, Changsha 410008, China; 4Health Management Center, Xiangya Hospital, Central South University, Changsha 410008, China; 5National Clinical Research Center for Geriatric Disorders, Xiangya Hospital, Central South University, Changsha 410008, China

**Keywords:** isovalerylcarnitine, docosapentaenoate, appendicular lean mass, grip strength, sarcopenia

## Abstract

Metabolites’ connection to sarcopenia through inflammation and mitochondrial dysfunction is presumed, but their impact remains unclear due to limitations in conventional observational studies caused by confounding bias and reverse causation. We conducted a Mendelian randomization (MR) analysis to elucidate the association of serum metabolites with sarcopenia and its related traits, i.e., appendicular lean mass and grip strength. Genetic instruments to proxy the serum metabolites were extracted from the most comprehensive genome-wide association study on the topic published so far. The corresponding summary statistics for the associations of genetic instruments with outcomes were calculated from the UK Biobank (n = 324,976 participants). The primary analyses were assessed by an inverse-variance weighted (IVW) method. The weighted median and MR-PRESSO methods were used as sensitive analyses. Fourteen genetically predicted serum metabolites were associated with the risk of sarcopenia (*P_IVW_* < 0.05). Two metabolites showed the overlapped association with sarcopenia and its related traits, which were isovalerylcarnitine (sarcopenia: odds ratio [OR] = 4.00, 95% confidence interval [CI] = 1.11~14.52, *P_IVW_* = 0.034; appendicular lean mass: β = −0.45 kg, 95% CI = −0.81~−0.09, *P_IVW_* = 0.015; grip strength: β = −1.51 kg, 95% CI = −2.31~−0.71, *P_IVW_* = 2.19 × 10^−4^) and docosapentaenoate (sarcopenia: OR = 0.16, 95% CI = 0.03~0.83, *P_IVW_* = 0.029; appendicular lean mass: β = −0.45 kg, 95% CI = 0.08~0.81, *P_IVW_* = 0.016). Twenty-seven metabolites were suggestive associated with appendicular lean mass or grip strength. This MR study provided evidence for the potential effects of metabolites on sarcopenia.

## 1. Introduction 

Sarcopenia is defined as a multifaceted systemic skeletal muscle disorder characterized by age-related declines in muscle mass and function [1]. Previous research consistently demonstrates an association between sarcopenia and heightened adverse outcomes, including frailty, disability, and mortality [2]. In a social context characterized by a constantly aging population, sarcopenia is garnering increasing attention [3]. However, to date, the pathophysiological mechanisms of sarcopenia have not been fully elucidated, impeding the development of targeted treatment strategies for managing this debilitating condition [4].

Accumulating evidences have suggested that metabolic factors, particularly those related to nutrition, play a role in the mechanisms of sarcopenia [5]. For example, reduced serum levels of docosahexaenoate and eicosapentaenoate, two omega-3 polyunsaturated fatty acids (PUFAs) renowned for their anti-inflammatory properties [6], have been reported to be associated with sarcopenia [7]. Docosapentaenoate might possess more potent anti-inflammatory properties compared to docosahexaenoate and eicosapentaenoate [8,9]; however, the relationships between docosapentaenoate and sarcopenia remain unexplored [10]. Isovalerylcarnitine, an acylcarnitine utilized for mitochondrial fatty acid β-oxidation [11], has been found to be marginally reduced in patients with sarcopenia [12]. Nevertheless, these findings were obtained from observational studies and thus may be susceptible to either confounding bias or reverse causation due to the conventional study design [13]. Further comprehensive research is required to build upon previous findings and acquire unbiased estimates of the causal links between metabolites and sarcopenia. This will offer fresh insights into the mechanisms of sarcopenia and potential targets for preventive and therapeutic strategies.

Mendelian randomization (MR) is an established genetic epidemiology method that employs genetic variants as proxies for the exposure of interest to evaluate the causal relationships between exposures and disease outcomes [14]. This approach offers distinct advantages over conventional observational studies. Firstly, the MR strategy can be free from conventional confounding by exploiting the random independent assortment of DNA during allelic meiotic divisions. Secondly, reverse causality bias would not occur due to the immutable nature of the transmitted germline genome [15]. Thirdly, in most cases, genetic variants are typically precisely measured and reported, and are less susceptible to bias and measurement errors. Thus, it is especially useful in evaluating risk factors of long-term effects [16]. In this context, we conducted a two-sample MR analysis to provide a comprehensive view of the relationships of serum metabolites with sarcopenia and its related traits, i.e., appendicular lean mass and grip strength in humans to identify metabolites that might contribute to the development of sarcopenia.

## 2. Methods

### 2.1. Study Design

The MR approach includes three fundamental assumptions: (1) ensuring strong associations between the instrumental variables and the exposures (relevance assumption), (2) guaranteeing that the instrumental variables remain unaffected by potential confounding factors that might impact the association between the exposures and outcomes (independence assumption), and (3) confirming the independence of instrumental variables from the outcome, given the specific exposure of interest (exclusion restriction assumption) [17]. We conducted a two-sample MR study to explore the association between serum metabolites and sarcopenia. Summary statistics (i.e., β coefficients and standard errors) of the associations between single nucleotide polymorphisms (SNPs) and serum metabolites were extracted from a large published genome-wide association study (GWAS) conducted in individuals of European descent [18]. Corresponding summary statistics for the associations of the serum metabolites-associated SNPs with sarcopenia, appendicular lean mass, and grip strength were calculated from the UK Biobank [19]. Detailed information on GWAS summary statistics or individual-level genetic data used in our analysis was presented in Appendix A.

### 2.2. Data on Genetic Instruments for Exposures

We obtained the instrumental variables for serum metabolites from an extensive GWAS meta-analysis involving 7824 individuals from two European population studies [18]. This comprehensive GWAS meta-analysis explored the genetic underpinnings of over 400 blood metabolites, employing advanced techniques including ultra-high-performance liquid-phase chromatography, gas-chromatography separations, and tandem mass spectrometry. After rigorous quality control procedures, 486 metabolites were chosen for genetic analysis, consisting of 309 known and 177 unknown metabolites. These 309 known metabolites were further categorized into 8 broad metabolic pathways, as outlined in the KEGG (Kyoto Encyclopedia of Genes and Genomes) database. These pathways encompass amino acids, carbohydrates, cofactors and vitamins, energy metabolism, lipids, nucleotides, peptides, and xenobiotic metabolism [20].

We used a relaxed statistical threshold of *p* < 1 × 10^−5^ to identify significant SNPs for each metabolite. This less-stringent threshold was usually used in MR analysis to obtain a larger variation for exposures when few genome-wide significant SNPs were available [21]. We incorporated all significant SNPs and performed a clumping procedure to eliminate interdependent SNPs with a pairwise linkage disequilibrium r^2^ < 0.01 within a 500-kb window, using the European 1000G reference panel as the basis. We excluded SNPs with the least significant associations with the exposure or those that were not present in the outcome datasets. Detailed information on selected SNPs was summarized in Appendix A.

### 2.3. Data on Genetic Associations with Outcomes

The primary outcomes of the study consisted of sarcopenia and its related traits (i.e., appendicular lean mass and grip strength). The relevant summary statistics concerning serum metabolite-associated SNPs in relation to sarcopenia and its indices were calculated from the UK biobank study, using logistic or linear regression models. Age, sex, genotype measurement batch, and 20 genetic principal components were included as covariates in regression analysis to address the potential population substructure. The UK biobank is a prospective population-based cohort established to allow detailed investigations of the genetic and nongenetic determinants of the diseases in middle and old age [19]. Over 500,000 participants aged 40 to 69 were drawn from 22 assessment centers across the United Kingdom between 2006 and 2010 [22]. All participants completed questionnaires on health and lifestyle factors during the baseline survey and provided blood samples for biomarker and genetic assays. We excluded participants who were non-white European ancestry (to minimize confounding by ancestry), sex mismatches, excess heterozygosity, missingness, or closer than 3rd-degree relatives in the present MR analysis. The definition of sarcopenia followed the criteria established by the European Working Group on Sarcopenia in Older People in 2019 [2]. Appendicular lean mass and grip strength were measured using bioelectrical impedance analysis and Jamar handheld dynamometer, respectively. This research has been conducted using the UK Biobank Resource under Application Number 77,646.

### 2.4. Statistical Analysis

The multiplicative random-effects inverse variance weighted (IVW) approach was applied as the main approach to calculate the estimates (βs) and the 95% confidence intervals (95% CIs) for each one-unit increase in serum metabolites. This method provided the most precise and unbiased estimates assuming that all SNPs were valid instrumental variables or horizontal pleiotropy was balanced [22].

Horizontal pleiotropy occurs when instrumental variables affect traits beyond the exposure pathway and directly influence the outcome, which might distort MR tests, leading to inaccurate causal estimates [23]. We assessed the potential horizontal pleiotropy using two additional MR sensitivity analyses: the weighted median method and the MR pleiotropy residual sum and outlier (MR-PRESSO) approach. These methods operate under distinct assumptions. The weighted median can provide consistent estimates as long as no less than 50% of the weight in the analysis comes from valid instrumental variables [24]. The MR-PRESSO method was further utilized to identify and remove outlier variants to correct the potential directional horizontal pleiotropy and address the detected heterogeneity [23]. Briefly, the global test can provide an estimate for the degree of horizontal pleiotropy (significant pleiotropy indicated by *p* < 0.05). The outlier corrected causal estimate can provide a corrected estimate for any significant pleiotropy detected, and the “distortion test” can provide an estimate for the degree to which the original and corrected estimates differ (*p* < 0.05 indicating a significant difference following corrections for pleiotropy). Tests two and three were implemented only in cases where *p* < 0.05 for global test estimates. Hence, the consistency of effects observed across multiple sensitivity analyses enhances the strength of the causal evidence [25].

Additionally, we used the F statistic and R^2^ to assess the strength and robustness of genetic instruments. The strength of each instrument was measured by calculating the F-statistic using the following formula: (F = beta^2^/SE^2^), where beta is the estimated genetic effect on serum metabolites, SE is the standard error of the genetic effect. An F statistic >10 was considered as a sufficiently strong instrument with a low potential for instrument bias [26]. R^2^ is the proportion of the variability of the serum metabolites explained by each instrument used the following formula: 2 × EAF × (1 − EAF) × beta^2^, where EAF is the effect allele frequency. To address the possibility of reverse causality relationship, we performed bidirectional MR analyses with metabolites as outcomes. We selected SNPs from GWAS that were significantly associated with appendicular lean mass and grip strength as the instrumental variables due to the absence of appropriate instrumental variables for sarcopenia [25]. Detailed information on selected SNPs was summarized in Appendix A.

We used the odds ratios (ORs) or betas (βs) and their 95% confidence intervals (CIs) as the estimates for the association of serum metabolite with the risk of sarcopenia or its related indices, respectively. To address the issue of multiple testing, the multiple-testing-adjusted threshold of the Bonferroni correction was applied to correct for the number of exposures tested for each metabolite and the significance of the causal feature was set to *p* < 1.03 × 10^−4^ (0.05/486). Serum metabolites were considered as causal features if they reached the Bonferroni adjusted significant level of *p* < 1.03 × 10^−4^. Bonferroni’s correction provides a straightforward approach to controlling the type I error rate by dividing the critical level of significance by the number of significance tests performed. However, the correction tends to be conservative if a large number of tests are performed [27]. Therefore, we defined serum metabolites as significant features if *p* < 0.05 for all three MR analyses (i.e., IVW, weighted median, and MR-PRESSO) and defined them as suggestive features if *P_IVW_* < 0.05 and either *P_weighted median_* < 0.05 or *P_MR-PRESSO_* < 0.05. *p*-values obtained are two-tailed for all statistical tests. Analyses were conducted in R software (version 4.1.2, R Foundation for Statistical Computing, Vienna, Austria).

## 3. Results

### 3.1. Study Overview

The analysis included 324,976 individuals of European descent with a mean age of 56.4 years and 46.4% being male. Among them, 561 (0.2%) individuals were diagnosed with sarcopenia. Table 1 displays the characteristics of the included individuals. In this study, we only presented the results of the known metabolites. The count of SNPs utilized as instrumental variables for each serum metabolites ranged from 3 to 226, with a median of 16. The instrumental variables explained 1.0–82.3% of the variance in corresponding serum metabolites. The minimum F statistic for validity tests of these genetic predictors was 17.64. All instrumental variables for the included serum metabolites were suitably robust (F statistic > 10) for MR analyses. The IVW primary analysis revealed that 43 serum metabolites were associated with sarcopenia or its related traits (*P_IVW_* < 0.05). These metabolites encompassed 11 amino acids, two carbohydrates, two cofactors and vitamins, one energy metabolite, 24 lipids, one nucleotide, and two peptides. And one serum metabolite remained causally associated with grip strength after Bonferroni correction.

### 3.2. Association of Serum Metabolites with Sarcopenia

Despite the Bonferroni correction yielding no significant causal associations between serum metabolites and sarcopenia, 14 genetically predicted serum metabolites exhibited an association with sarcopenia risk at the normal significance level of 0.05 (see Table 2). The percentage of variation of the metabolite explained by the genetic instrumental SNPs is indicated in the column Variance. A suggestive association with sarcopenia was identified for 11 serum metabolites (with *P_IVW_* < 0.05 and either *P_weighted median_* < 0.05 or *P_MR-PRESSO_* < 0.05) (refer to Appendix A). Specifically, the findings provided robust evidence supporting a significant association between valerate and 5α-pregnan-3β,20α-disulfate with sarcopenia (*p* < 0.05 for all three MR analyses). Each one-unit increase in genetically predicted valerate and 5α-pregnan-3β,20α-disulfate was associated with a lower risk of sarcopenia with OR (95% CI) ranging from 0.01 (0.00, 0.89), *P_IVW_* = 1.37 × 10^−4^ to 0.42 (0.20, 0.87), *P_IVW_* = 0.019. Effect estimates remained broadly consistent across the other two MR sensitivity methods (i.e., weighted median and MR-PRESSO), suggesting the robustness of the associations between these two serum metabolites and the risk of sarcopenia. The global test of MR-PRESSO performed to rule out the possibility of horizontal pleiotropy is shown in Appendix A.

It is worth noting that isovalerylcarnitine and docosapentaenoate were not only associated with the risk of sarcopenia, but also associated with its related traits (i.e., appendicular lean mass and grip strength). For example, in alignment with the findings indicating an increased risk of sarcopenia (OR = 4.00, 95% CI: 1.11 to 14.52, *P_IVW_* = 0.034), genetic predicted isovalerylcarnitine was also significantly associated with decreased grip strength (β = −0.45 kg, 95% CI: −1.13 to −0.13, *P_IVW_* = 0.014) and reduced appendicular lean mass (β = −0.22 kg, 95% CI: −0.44 to −0.01, *P_IVW_* = 0.048). Genetic predicted docosapentaenoate was notably associated with a reduced risk of sarcopenia, and it was also significantly correlated with increased appendicular lean mass (β = 0.45 kg, 95% CI: 0.08 to 0.081, *P_IVW_* = 0.016). Figure 1 illustrates the metabolites that overlapped in their associations with sarcopenia and its related traits.

### 3.3. Association of Serum Metabolites with Appendicular Lean Mass

The effects of genetically predicted serum metabolites on the level of appendicular lean mass are presented in Table 3. Using the IVW MR method, 22 genetically predicted serum metabolites were associated with the appendicular lean mass at the significant level of 0.05. No serum metabolite remained significant after the Bonferroni correction. Except for nonanoylcarnitine and octadecanedioate, all serum metabolites were identified to be suggestive associated with appendicular lean mass. Furthermore, we reported eight features that passed all sensitivity analyses without horizontal pleiotropy (Table 3 and Appendix A). The direction of these genetically predicted serum metabolites was consistent across the three MR methods tested. The increase in the appendicular lean mass could be attributed to the increase in the mannose (*P_IVW_* = 0.002, *P_Weighted-Median_* = 0.006, *P_MR-PRESSO_* = 0.005), glycine (*P_IVW_* = 0.002, *P_Weighted-Median_* = 8.53 × 10^−16^, *P_MR-PRESSO_* = 0.005), docosapentaenoate (*P_IVW_* = 0.016, *P_Weighted-Median_* = 0.028, *P_MR-PRESSO_* = 0.026), 1-arachidonoyl-glycerophosphocholine (*P_IVW_* = 0.008, *P_Weighted-Median_* = 0.002, *P_MR-PRESSO_* = 0.016), and 5α-Adiol disulfate (*P_IVW_* = 0.028, *P_Weighted-Median_* = 0.047, *P_MR-PRESSO_* = 0.042). Each one-unit increase in genetically predicted hexanoylcarnitine (*P_IVW_* = 0.016, *P_Weighted-Median_* = 0.015, *P_MR-PRESSO_* = 0.027), o-methylascorbate (*P_IVW_* = 0.003, *P_Weighted-Median_* = 7.81 × 10^−6^, and *P_MR-PRESSO_* = 0.005), and decanoylcarnitine (*P_IVW_* = 0.002, *P_Weighted-Median_* = 0.024, *P_MR-PRESSO_* = 0.008) were associated with decreased appendicular lean mass. Sensitive results of the global test of MR-PRESSO are presented in Appendix A.

### 3.4. Association of Serum Metabolites with Grip Strength

The effects of genetically predicted serum metabolites on grip strength are presented in Table 4. The 3-dehydrocarnitine (*P_IVW_* = 9.19 × 10^−5^) remained significantly associated with grip strength after Bonferroni adjustment. A total of eleven genetically predicted serum metabolites were associated with grip strength at the nominal significance (*P_IVW_* < 0.05) and nine of them were identified to be suggestively associated with grip strength. The direction of effects of the following five serum metabolites (i.e., hyodeoxycholate, androsterone sulfate, glycine, 3-dehydrocarnitine, and epiandrosterone sulfate) were consistent across three MR methods, which showed strong evidence supporting the suggestive casual associations with grip strength. The increase in the grip strength could be attributed to the genetically predicted increase in the hyodeoxycholate (*P_IVW_* = 0.017, *P_Weighted-Median_* = 0.031, *P_MR-PRESSO_* = 0.023) and glycine (*P_IVW_* = 0.006, *P_Weighted-Median_* = 8.42 × 10^−5^, *P_MR-PRESSO_* = 0.011). However, IVW MR revealed evidence of a negative association of genetically predicted androsterone sulfate (β = −0.17 kg, 95% CI: −0.30 to −0.03, *P_IVW_* = 0.014), 3-dehydrocarnitine (β = −1.51 kg, 95% CI: −2.27 to −0.75, *P_IVW_* = 9.19 × 10^−5^), and epiandrosterone sulfate (β = −0.33 kg, 95% CI: −0.53 to −0.13, *P_IVW_* = 0.001) with grip strength. The results of the global test of MR-PRESSO are presented in Appendix A.

### 3.5. Complementary Analyses

We conducted bidirectional MR analyses to assess whether there is a reverse causal relationship of sarcopenic indices with the identified metabolites. Our MR results showed limit evidence for a causal effect of genetically predicted appendicular lean mass and grip strength on the identified metabolites (Appendix A), including isovalerylcarnitine (appendicular lean mass: β = −0.007, 95% CI: −0.027 to 0.013, *p* = 0.498; grip strength: β = −0.034, 95% CI: −0.095 to 0.027, *p* = 0.273) and docosapentaenoate (appendicular lean mass: β = −0.011, 95% CI: −0.033 to 0.011, *p* = 0.279; grip strength: β = −0.013, 95% CI: −0.076 to 0.050, *p* = 0.693).

## 4. Discussion

We conducted an MR study to provide unbiased detection of the relationships between human serum metabolites and sarcopenia, as well as their impact on appendicular lean mass and grip strength variations. Among 14 metabolites identified as significantly associated with sarcopenia using the IVW method, 11 metabolites showed significant associations in sensitivity analyses using the weighted median or MR-PRESSO method. Furthermore, we found that isovalerylcarnitine might contribute to the risk of sarcopenia and adversely affect both lean mass and grip strength. Conversely, docosapentaenoate may reduce the risk of sarcopenia by increasing appendicular lean mass. Robust associations of these two metabolites and sarcopenia were indicated as all estimates from sensitivity analyses were directionally concordant. Moreover, bidirectional analyses showed no potential effect of sarcopenic indices on these metabolites, suggesting that there was no reverse causality.

### 4.1. Comparison with Previous Studies

Some observational studies have investigated the link between certain metabolites (e.g., isovalerylcarnitine and docosapentaenoate) and sarcopenia. A cross-sectional study of 65 men with sarcopenia and 181 men without sarcopenia found no significant association between isovalerylcarnitine and sarcopenia in multivariate analysis [12], whereas our study found that isovalerylcarnitine was associated with the increased risk of sarcopenia as well as a decreased level of appendicular lean mass and grip strength. This disparity between previous and current research may stem from variations in research designs. Observational studies are susceptible to confounding, reverse causation, and other biases, which can lead to incorrect causal inference, highlighting the importance of methods such as MR for estimating the magnitude of the direct causal effect unbiasedly and disentangling such scenarios [28].

Omega-3 PUFAs, including docosahexaenoate, eicosapentaenoate, and docosapentaenoate, are a class of long-chain fatty acids and may be a potential therapy or preventive measure for sarcopenia according to previous randomized controlled trials [29,30]. Nevertheless, the independent effects of docosahexaenoate, eicosapentaenoate, and docosapentaenoate on sarcopenia have not undergone comprehensive investigation. An observational study showed that docosahexaenoate and eicosapentaenoate were inversely associated with sarcopenia diagnosed solely by skeletal muscle mass [7]. While another cross-sectional study observed no significant differences of prevalence of sarcopenia between the quartiles of docosahexaenoate and eicosapentaenoate in adjusted models [31]. Additionally, no study has yet explored the relationship between docosapentaenoate and sarcopenia. In this study, we identified that docosapentaenoate has relationships with the decreased risk of sarcopenia and the increased level of appendicular lean mass, but docosahexaenoate and eicosapentaenoate are not significantly associated with sarcopenia and its components. These results indicate that docosapentaenoate might be a potent ingredient in omega-3 PUFAs in terms of preventing and treating sarcopenia.

### 4.2. Possible Explanations

Isovalerylcarnitine belongs to the group of acylcarnitines and is generated during leucine catabolism [32]. Isovalerylcarnitine has been proved to be able to activate the calpain system, producing an early and marked increase in apoptosis and cell killing [33]. As the impaired skeletal muscle regeneration capacity in old age, increased apoptosis may eventually lead to sarcopenia [34]. In addition, it is known that a dysregulated catabolic metabolism in the mitochondria can lead to the accumulation of acylcarnitines, and then could contribute to a positive feedback loop, causing elevated production of reactive oxygen species and further mitochondrial derangement [35,36]. So it can be inferred that the effect of isovalerylcarnitine on bioenergetic dysfunction in muscle tissue could eventually lead to sarcopenia.

Docosapentaenoate, as well as the well-known essential nutrients docosahexaenoate and eicosapentaenoate, offers benefits for numerous aging processes and has demonstrated more favorable effects on metabolic disorders than docosahexaenoate and eicosapentaenoate [9,37]. Docosapentaenoate and its derived pro-resolving lipid mediators have been found having protective effects against cardiometabolic disease [38]. Furthermore, compared with docosahexaenoate and eicosapentaenoate, docosapentaenoate supplement can improve the homeostasis model assessment of insulin resistance more significantly [39]. The favorable effects of docosapentaenoate on metabolism may contribute to the prevention of metabolism-related diseases, including sarcopenia. Additionally, docosapentaenoate has been proved to be potent in inhibiting the pro-inflammatory signaling pathways [38]. Docosapentaenoate can be metabolized into docosapentaenoate-derived protectin (PDn-3DPA) and docosapentaenoate-derived D-series resolvins (RvDn-3DPA), which were considered to have an anti-inflammation function [9] PDn-3DPA has been identified as playing a crucial role in regulating macrophage resolution responses, Ref. [40] and RvDn-3DPA can reduce leukocyte and platelet activation in peripheral blood [41]. Moreover, docosapentaenoate proved to be the most potent inhibitor of COX-1 activity, with docosahexaenoate and eicosapentaenoate having a weaker effect [8]. Considering the systemic chronic low-grade inflammation in sarcopenia, docosapentaenoate may have a protective effect against muscle loss via itself and its derivatives.

### 4.3. Strengths and Limitations

Our study has several strengths. First, to the best of our knowledge, this is the first MR study to evaluate the relationships of serum metabolites on sarcopenia. Such a design can mitigate limitations associated with confounding factors often encountered in conventional observational studies and can offer more robust evidence of causality between exposure and outcome. Second, utilizing the most extensive and current GWAS available for metabolites enabled the establishment of a robust instrumental variable, predicted to yield unbiased MR estimates. Third, we confined the study population to individuals of European descent, which reduced the bias due to population stratification. Fourth, we conducted several sensitivity analyses (i.e., weighted median and MR-PRESSO methods), which can control the bias caused by pleiotropic effects to validate the associations observed in the IVW analysis.

There are several limitations of our study. First, although we have used MR approaches to guard against confounding, we nevertheless cannot fully exclude residual bias due to unmeasured confounders, which is an established limitation of the MR approach [42]. Second, as the number of sarcopenia cases in our study was relatively small, we could not completely rule out the possibility that our study may not have adequate power to discover a weak association. Third, while this study suggests that some metabolites are associated with sarcopenia risk, it primarily provides a prediction without verification. The causal associations and molecular mechanisms still need to be more thoroughly investigated and confirmed in future studies.

### 4.4. Clinical and Research Implications

Sarcopenia is one of the most important causes of frailty and is associated with higher medical costs. However, owing to an incomplete understanding of the underlying mechanisms and a failure to intervene early enough in the pathological cascade, no specific drugs have been approved for the treatment of sarcopenia [4]. Our study expanded previous observational findings and additionally assessed the relationships of metabolites with sarcopenia. These findings may help to reveal the impact of metabolites on the development of muscle loss and open new paths for the pathogenetic studies of sarcopenia. We anticipate that our study results could serve as a robust foundation and identify potential specific targets for interventions involving metabolites, aimed at preventing and treating sarcopenia.

## 5. Conclusions

Through this large-scale MR study, we investigated the effects of 309 serum metabolites on sarcopenia and the variations of appendicular lean mass and grip strength. Fourteen metabolites were found to have potential associations with sarcopenia, including isovalerylcarnitine and docosapentaenoate, which overlapped in both sarcopenia and its related traits. Our study may provide novel insights into the pathological mechanisms and the development of treatment in this field. Further studies are warranted to validate our results and elucidate the potential molecular mechanisms linking these metabolites to sarcopenia.

## Figures and Tables

**Figure 1 nutrients-15-03964-f001:**
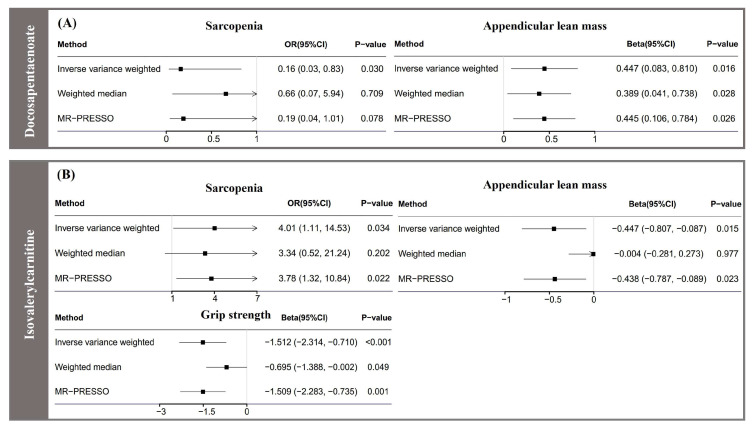
Forest plot for association of isovalerylcarnitine and docosapentaenoate with sarcopenia and its related traits. CI, confidence interval; MR-PRESSO, Mendelian randomization pleiotropy residual sum and outlier; OR, odds ratio. (**A**) Forest plot for association of docosapentaenoate with sarcopenia and its related trait; (**B**) Forest plot for association of isovalerylcarnitine with sarcopenia and its related traits.

**Table 1 nutrients-15-03964-t001:** Participant characteristics at recruitment.

Characteristics	
All subjects at recruitment, n	324,976
Age, mean (SD), years	56.4 (8.1)
Body mass index, mean (SD), kg/m^2^	27.4 (4.8)
Female sex, n (%)	174,139 (53.6)
Sarcopenia, n (%)	561 (0.2)
Grip strength, mean (SD), kg	22.4 (5.3)
Appendicular lean mass, mean (SD), kg	32.9 (11.3)

SD, standard deviation; n, number of participants.

**Table 2 nutrients-15-03964-t002:** Summary of association between lead metabolites and sarcopenia risk by using IVW method.

Metabolites	N (SNPs)	Variance, %	OR (95% CI)	*p*
4-acetamidobutanoate *	35	16.3	0.10 (0.01, 0.84)	0.034
Biliverdin *	19	26.1	2.49 (1.06, 5.82)	0.036
Taurocholate	16	21.4	1.52 (1.03, 2.24)	0.037
Myo-inositol *	40	18.6	0.16 (0.03, 0.93)	0.042
Phenyllactate *	19	11.3	0.13 (0.02, 0.81)	0.029
Levulinate (4-oxovalerate) *	61	22.7	5.50 (1.12, 26.89)	0.035
Acetylcarnitine *	21	10.3	10.83 (1.55, 76.81)	0.016
Docosapentaenoate (n3)	11	7.3	0.16 (0.03, 0.83)	0.029
5α-pregnan-3β,20α-disulfate **	15	15.3	0.42 (0.20, 0.87)	0.019
Gamma-glutamylmethionine	9	10.8	7.61 (1.07, 53.99)	0.042
Valerate **	10	6.0	0.01 (0.00, 0.89)	<0.001
Isovalerylcarnitine *	21	13.5	4.00 (1.11, 14.52)	0.034
1-arachidonoylglycerophosphoethanolamine *	26	11.7	0.19 (0.04, 0.95)	0.042
Hydroxyisovaleroyl carnitine *	8	6.4	10.96 (1.65, 72.86)	0.013

* where *P_IVW_* < 0.05 and *P_weighted median_*/*P_MR-PRESSO_* < 0.05, it means suggestive association. ** where *P_IVW_* < 0.05, *P_weighted median_* and *P_MR-PRESSO_* < 0.05, it means significant association. CI, confidence interval; IVW, inverse-variance weighted; OR, odds ratio; SNPs, single nucleotide polymorphisms.

**Table 3 nutrients-15-03964-t003:** Summary of association between lead metabolites and appendicular lean mass.

Metabolites	N (SNPs)	Variance, %	β (95% CI)	*p*
Mannose **	21	13.9	0.91 (0.34, 1.48)	0.002
Citrate *	46	15.8	0.46 (0.10, 0.83)	0.013
Guanosine *	13	13.0	−0.12 (−0.24, −0.00)	0.050
Betaine *	25	13.4	−0.90 (−1.50, −0.28)	0.004
Beta-hydroxyisovalerate *	23	9.9	−0.49 (−0.89, −0.09)	0.014
3-methylhistidine *	11	6.5	−0.14 (−0.25, −0.03)	0.016
Serine *	35	13.1	0.91 (0.08, 1.74)	0.031
Hexanoylcarnitine **	19	13.0	−0.40 (−0.72, −0.07)	0.016
Glycine **	26	18.7	0.51 (0.19, 0.84)	0.002
O-methylascorbate **	49	28.2	−0.44 (−0.74, −0.14)	0.003
Docosapentaenoate (n3) **	11	7.3	0.45 (0.08, 0.81)	0.016
1-arachidonoylglycerophosphocholine **	21	13.4	0.33 (0.08, 0.57)	0.008
Decanoylcarnitine **	15	9.9	−0.34 (−0.55, −0.12)	0.002
1-palmitoylglycerophosphocholine *	34	11.2	−0.48 (−0.92, −0.03)	0.037
1-oleoylglycerophosphocholine *	17	4.2	−0.37 (−0.65, −0.09)	0.010
10-heptadecenoate (17:1n7) *	6	2.5	0.40 (0.03, 0.78)	0.034
Isovalerylcarnitine *	21	13.5	−0.45 (−0.81, −0.09)	0.015
nonanoylcarnitine	16	15.7	0.13 (0.00, 0.26)	0.048
Tetradecanedioate *	20	19.4	0.13 (0.02, 0.25)	0.026
Octadecanedioate	10	6.3	0.32 (0.01, 0.63)	0.043
5α-Adiol disulfate **	18	12.2	0.14 (0.02, 0.27)	0.028
4α-Adiol disulfate 2 *	19	6.9	0.33 (0.06, 0.61)	0.018

* where *P_IVW_* < 0.05 and *P_weighted median_*/*P_MR-PRESSO_* < 0.05, it means suggestive association. ** where *P_IVW_* < 0.05, *P_weighted median_* and *P_MR-PRESSO_* < 0.05, it means strong evidence supported the suggestive association. CI, confidence interval; SNPs, single nucleotide polymorphisms.

**Table 4 nutrients-15-03964-t004:** Summary of association between lead metabolites and grip strength.

Metabolites	N (SNPs)	Variance, %	β (95% CI)	*p*
Laurate (12:0) *	41	16.0	−1.04 (−1.86, −0.21)	0.014
Gamma-glutamyltyrosine	47	17.3	0.76 (0.01, 1.51)	0.047
Glucose *	38	15.4	−0.80 (−1.59, −0.01)	0.046
1-oleoylglycerol (1-monoolein)	16	7.5	−0.54 (−1.05, −0.02)	0.040
Hyodeoxycholate **	17	12.5	0.29 (0.05, 0.52)	0.017
Androsterone sulfate **	22	20.7	−0.17 (−0.30, −0.03)	0.014
Glycine **	26	18.7	0.66 (0.19, 1.13)	0.006
3-dehydrocarnitine **	27	11.6	−1.51 (−2.27, −0.75)	<0.001
1-heptadecanoyl glycerophosphocholine *	10	7.3	1.10 (0.26, 1.93)	0.010
Epiandrosterone sulfate **	13	12.3	−0.33 (−0.53, −0.13)	0.001
Isovalerylcarnitine *	21	13.5	−1.51 (−2.31, −0.71)	<0.001

* where *P_IVW_* < 0.05 and *P_weighted median_*/*P_MR-PRESSO_* < 0.05, it means suggestive association. ** where *P_IVW_* < 0.05, *P_weighted median_* and *P_MR-PRESSO_* < 0.05, it means strong evidence supported the suggestive association. CI, confidence interval; SNPs, single nucleotide polymorphisms.

## Data Availability

Primary data from the UK Biobank resource are accessible upon application: https://www.ukbiobank.ac.uk/ (accessed on 10 August 2023). The complete Genome-wide association study summary-level data for serum metabolics are available at https://pubmed.ncbi.nlm.nih.gov/24816252/.

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
