# Peer review of "Genetically Predicted Levels of Serum Metabolites and Risk of Sarcopenia: A Mendelian Randomization Study"

_nutrients, 2023, doi:10.3390/nu15183964_

Round 1
Reviewer 1 Report
The manuscript titled "Genetically predicted levels of serum metabolites and risk of sarcopenia: a Mendelian randomization study" by Sha et al. investigated the relationship between serum metabolite levels and the risk of sarcopenia using two GWAS datasets. One dataset was used to predict serum metabolite levels, while the other was used to assess the association of these metabolite-related SNPs to sarcopenia. The authors discovered two predicted serum metabolite levels that were significantly associated with sarcopenia, appendicular lean mass, and grip strength. Although findings of serum metabolite associated with sarcopenia could aid in the development of therapeutic treatments and disease control, I have some severe concerns about the manuscript.
1. Simply testing genetic variations is insufficient to determine the “causal” relationships between serum metabolites and diseases or phenotypes. The authors must provide evidence of these "causality" relationships.
2. In lines 166-169, the authors mention that each metabolite is linked to 3 to 226 SNPs, and these SNPs can explain 1-82% of the variance. Is the explanation cumulative, and what is the amount of variation explained by these SNPs in significant serum metabolites? Without this information, it is difficult to fully comprehend the study's implications.
3. No supplemental materials were attached.
4. The current study design description is difficult to understand. I recommend that the authors enhance the method section for better comprehension. This will also aid other researchers in reproducing the study.
Minor concerns:
The Supplementary Materials and Data Availability Statement sections should be revised.
Moderate editing of English language required
Author Response
Response to Reviewer 1 Comments
|
||
1. Summary |
|
|
Thank you very much for taking the time to review this manuscript. Please find the detailed responses below and the corresponding revisions in the re-submitted files.
|
||
2. Questions for General Evaluation |
Reviewer’s Evaluation |
Response and Revisions |
Does the introduction provide sufficient background and include all relevant references? |
Can be improved |
Thank you very much for your suggestions. We have made corresponding revisions to each of your question, please review our point-by-point responses below and re-submitted files. |
Are all the cited references relevant to the research? |
Can be improved |
|
Is the research design appropriate? |
Must be improved |
|
Are the methods adequately described? |
Must be improved |
|
Are the results clearly presented? |
Must be improved |
|
Are the conclusions supported by the results? |
Must be improved |
|
3. Point-by-point response to Comments and Suggestions for Authors |
||
Major comment: The manuscript titled “Genetically predicted levels of serum metabolites and risk of sarcopenia: a Mendelian randomization study” by Sha et al. investigated the relationship between serum metabolite levels and the risk of sarcopenia using two GWAS datasets. One dataset was used to predict serum metabolite levels, while the other was used to assess the association of these metabolite-related SNPs to sarcopenia. The authors discovered two predicted serum metabolite levels that were significantly associated with sarcopenia, appendicular lean mass, and grip strength. Although findings of serum metabolite associated with sarcopenia could aid in the development of therapeutic treatments and disease control, I have some severe concerns about the manuscript. |
||
Response: We greatly appreciate your comments and have made corresponding revisions and responses to each comment below. |
||
Comments 1: Simply testing genetic variations is insufficient to determine the “causal” relationships between serum metabolites and diseases or phenotypes. The authors must provide evidence of these “causality” relationships. |
||
Response: Many thanks for your comment. Using genetic variations for detecting causal relationships may indeed be insufficient. Our findings suggest a potential relationship between serum metabolites and sarcopenia, but further research is still needed to validate these findings and provide evidence supporting causality relationships. We have addressed the limitation in the manuscript and modified our interpretation of the results to make it more suitable. Mendelian randomization can still be a statistical method that can effectively suggest causal inference between phenotypes. In Mendelian randomization, individuals within a population can be stratified into subgroups based on their genetic variations. Genetic variation is independent of environmental and other variables, and the genetic code for each individual is determined before birth. Therefore, in effect, we are conducting a natural trial within a population where nature has randomly given some individuals a genetic “treatment” which elevate their exposure levels. (Stephen Burgess, et al. Mendelian Randomization: Methods for Causal Inference Using Genetic Variants. 2021. ISBN:9780367341848) Furthermore, in this revision, we performed bidirectional MR analyses with the metabolites previously found to be significantly associated with sarcopenia or its indicators as outcomes, addressing the potential for reverse causality. Bidirectional analyses showed no potential effect of appendicular lean mass and grip strength on these metabolites, further emphasizing that the significant relationships were from metabolites to sarcopenia, rather than the other way around. Action: “To address the possibility of reverse causality relationship, we performed bidirectional MR analyses with metabolites as outcomes. We selected SNPs from GWAS which were significantly associated with appendicular lean mass and grip strength as the instrumental variables due to the absence of appropriate instrumental variables for sarcopenia.[28] Detailed information on selected SNPs was summarized in Supplementary Table 6.” (Pages 4, lines 160, in the clean copy of the revised manuscript) “3.5. Complementary analyses We conducted bidirectional MR analyses to assess whether there is a reverse causal relationship of sarcopenic indices with the identified metabolites. Our MR results showed limit evidence for a causal effect of genetically predicted appendicular lean mass and grip strength on the identified metabolites(Supplementary table 7), including isovalerylcarnitine (appendicular lean mass: β=-0.007, 95% CI: -0.027 to 0.013, P=0.498; grip strength: β=-0.034, 95% CI: -0.095 to 0.027, P=0.273) and docosapentaenoate (appendicular lean mass: β=-0.011, 95% CI: -0.033 to 0.011, P=0.279; grip strength: β=-0.013, 95% CI: -0.076 to 0.050, P=0.693). ” (Pages 8, lines 278, in the clean copy of the revised manuscript) “Third, while this study suggests that some metabolites are associated with sarcopenia risk, it primarily provides a prediction without verification. The causal associations and molecular mechanisms still need to be more thoroughly investigated and confirmed in future studies.” (Pages 10, lines 378, in the clean copy of the revised manuscript) “Through this large-scale MR study, we investigated the effects of 309 serum metabolites on sarcopenia and the variations of appendicular lean mass and grip strength. Fourteen metabolites were found to have potential associations with sarcopenia, including isovalerylcarnitine and docosapentaenoate overlapped in both sarcopenia and its related traits. Our study may provide novel insights into the pathological mechanisms and the development of treatment in this field. Further studies are warranted to validate our results and elucidate the potential molecular mechanisms between these metabolites and sarcopenia.” (Pages 10, lines 394, in the clean copy of the revised manuscript) |
||
Comments 2: In lines 166-169, the authors mention that each metabolite is linked to 3 to 226 SNPs, and these SNPs can explain 1-82% of the variance. Is the explanation cumulative, and what is the amount of variation explained by these SNPs in significant serum metabolites? Without this information, it is difficult to fully comprehend the study's implications. |
||
Response: Many thanks for your comment. The explanation represents the cumulative effect of all instrumental variables specific to a particular exposure on the variation in that exposure. We used the R2 to assess the strength and robustness of genetic instruments. R2 is the proportion of the variability of the serum metabolites explained by each instrument. To enhance the comprehension of R2, we have added a definition for R2 and a formula to explain how the variance is calculated in the revised manuscript. Per your suggestion, we have provided the amount of variation explained by genetic instrumental SNPs in significant serum metabolites in Tables 2-4 of the revised manuscript. Action: “Additionally, we used the F statistic and R2 to assess the strength and robustness of genetic instruments. The strength of each instrument was measured by calculating the F-statistic using the following formula: (F=beta2/SE2), where beta is the estimated genetic effect on serum metabolites, and SE is the standard error of the genetic effect. An F statistic >10 was considered a sufficiently strong instrument with a low potential for instrument bias. R2 is the proportion of the variability of the serum metabolites explained by each instrument using the following formula: 2*EAF*(1-EAF) *beta2, where EAF is the effect allele frequency.” (Pages 4, lines 153, in the clean copy of the revised manuscript) Please see the updated Tables 2-4 in the revised manuscript. |
||
Comments 3: No supplemental materials were attached. |
||
Response: Many thanks for your comment. We have ensured that the Supplemental Materials can be viewed now. Action: Please see the Supplemental Materials. |
||
Comments 4: The current study design description is difficult to understand. I recommend that the authors enhance the method section for better comprehension. This will also aid other researchers in reproducing the study. |
||
Response: Thanks for your valuable comment. Per your suggestion, we have revised the Methods and Results sections of the revised manuscript for better comprehension and readability. Specifically, we have further elaborated on the details of the Mendelian randomization method in the Methods section, added a comprehensive explanation of the reliability validation for instrumental variables and an introduction to Bonferroni correction. In addition, we have adjusted the wording to enhance the overall clarity of the article. Action: Please see the updated Methods and Results sections in the revised manuscript. |
||
Minor concerns: The Supplementary Materials and Data Availability Statement sections should be revised. |
||
Response: Many thanks for your comment. We have revised the Supplementary Materials and Data Availability Statement sections. Action: Please see the Supplemental Materials and Data Availability Statement in the revised manuscript. |
||
Response to Comments on the Quality of English Language: Moderate editing of the English language required |
||
Response: We appreciate your comment. We have edited and polished the language throughout the entire manuscript. Action: Please see the revised manuscript. |

Reviewer 2 Report
In the manuscript # nutrients-2577942 by Sha et al., "Genetically predicted levels of serum metabolites and risk of 2 sarcopenia: a Mendelian randomization study", authors provided optimal evidence for the potentially causal effects of metabolites on sarcopenia based on a large population and convincing research methods. It is scientifically sound and contains sufficient interest and originality to merit publication. Although I have no serious criticisms regarding methodology, results and interpretation of results, it would be clearer to the reader if the authors discussed these results using the weighted median and MR-PRESSO methods in comparison to the inverse variance weighting (IVW) method in Discussion section.
Author Response
Response to Reviewer 2 Comments
|
||
1. Summary |
|
|
Thank you very much for taking the time to review this manuscript. Please find the detailed responses below and the corresponding revisions in the re-submitted files.
|
||
2. Questions for General Evaluation |
Reviewer’s Evaluation |
Response and Revisions |
Does the introduction provide sufficient background and include all relevant references? |
Yes |
Many thanks for your positive evaluation. |
Are all the cited references relevant to the research? |
Yes |
|
Is the research design appropriate? |
Yes |
|
Are the methods adequately described? |
Yes |
|
Are the results clearly presented? |
Yes |
|
Are the conclusions supported by the results? |
Yes |
|
3. Point-by-point response to Comments and Suggestions for Authors |
||
Major comment: In the manuscript # nutrients-2577942 by Sha et al., “Genetically predicted levels of serum metabolites and risk of 2 sarcopenia: a Mendelian randomization study”, authors provided optimal evidence for the potentially causal effects of metabolites on sarcopenia based on a large population and convincing research methods. It is scientifically sound and contains sufficient interest and originality to merit publication. Although I have no serious criticisms regarding methodology, results and interpretation of results, it would be clearer to the reader if the authors discussed these results using the weighted median and MR-PRESSO methods in comparison to the inverse variance weighting (IVW) method in Discussion section. |
||
Response: Many thanks for your positive comments. We have discussed these results using the weighted median and MR-PRESSO methods in comparison to the IVW method in Discussion section. Action: “Among 14 metabolites identified as significantly associated with sarcopenia using the IVW method, 11 metabolites showed significant associations in sensitivity analyses using the weighted median or the MR-PRESSO method. Furthermore, two of metabolites, i.e., isovalerylcarnitine and docosapentaenoate had been replicated with a consistent direction and a significant PIVW<0.05 in sarcopenic indices, i.e., appendicular lean mass or grip strength. Robust associations of these two metabolites and sarcopenia were indicated as all estimates from sensitivity analyses were directionally concordant.” (Pages 9, lines 325, in the clean copy of the revised manuscript) “Fourth, we conducted several sensitivity analyses (i.e., weighted median and MR-PRESSO methods), which can control the bias caused by pleiotropic effects to validate the associations observed in the IVW analysis.” (Pages 10, lines 369, in the clean copy of the revised manuscript) |
Reviewer 3 Report
The reviewed article presents a very interesting study regarding the link between genetically predicted levels of serum metabolites and risk of sarcopenia. In my opinion the study is important due to the frequency of the disorder and yet the lack of many relevant data.
It is a very nice designed and conducted study and the article is fluent and easy to follow.
Some remarks:
- since the method used is “Mendelian randomization” please expand the introductory part related to this tool
- row 110 “Details of this cohort are available online and published elsewhere” – it would be nice for the readers to have more details when they read the article instead of directly check the reference
- please include some references and explanations about “Bonferroni correction”
Author Response
Response to Reviewer 3 Comments
|
||
1. Summary |
|
|
Thank you very much for taking the time to review this manuscript. Please find the detailed responses below and the corresponding revisions in the re-submitted files.
|
||
2. Questions for General Evaluation |
Reviewer’s Evaluation |
Response and Revisions |
Does the introduction provide sufficient background and include all relevant references? |
Can be improved |
Thank you very much for your suggestions. We have made corresponding revisions to each of your question, please review our point-by-point responses below and re-submitted files. |
Are all the cited references relevant to the research? |
Can be improved |
|
Is the research design appropriate? |
Can be improved |
|
Are the methods adequately described? |
Yes |
|
Are the results clearly presented? |
Yes |
|
Are the conclusions supported by the results? |
Yes |
|
3. Point-by-point response to Comments and Suggestions for Authors |
||
Major comment: The reviewed article presents a very interesting study regarding the link between genetically predicted levels of serum metabolites and risk of sarcopenia. In my opinion the study is important due to the frequency of the disorder and yet the lack of many relevant data. It is a very nice designed and conducted study and the article is fluent and easy to follow. |
||
Response: We appreciate your positive comments. Please see our response to each comment below. |
||
Comments 1: since the method used is “Mendelian randomization” please expand the introductory part related to this tool. |
||
Response: We appreciate your comment and have provided a more detailed description of this method in the introduction section and method section of the revised manuscript. Action: “Mendelian randomization (MR) is an established genetic epidemiology method that employs genetic variants as proxies for the exposure of interest to evaluate the causal relationships between exposures and disease outcomes.[14] This approach offers distinct advantages over conventional observational studies. Firstly, the MR strategy can be free from conventional confounding by exploiting the random independent assortment of DNA during allelic meiotic divisions. Secondly, reverse causality bias would not occur due to genetic variation is independent of environmental variables and determined before birth.[15] Thirdly, in most cases, genetic variants are typically precisely measured and reported and are less susceptible to bias and measurement errors. Thus, it is especially useful in evaluating risk factors of long-term effects.[16]” (Pages 2, lines 60, in the clean copy of the revised manuscript) “We conducted a two-sample MR study using summary-level statistics from a public genome-wide association study (GWAS) conducted in European-descent individuals. The three fundamental assumptions of MR include: (1) ensuring strong associations between the instrumental variables and the exposures (relevance assumption), (2) guaranteeing that the instrumental variables remain unaffected by potential confounding factors that might impact the connection between the exposures and outcomes (independence assumption), and (3) confirming the independence of instrumental variables from the outcome, given the specific exposure of interest (exclusion restriction assumption).[18]” (Pages 2, lines 75, in the clean copy of the revised manuscript) |
||
Comments 2: row 110 “Details of this cohort are available online and published elsewhere” – it would be nice for the readers to have more details when they read the article instead of directly check the reference. |
||
Response: Thanks for your valuable comment. We have added more details about the UK Biobank in the manuscript. Action: “The primary outcomes of the study consisted of sarcopenia and its related traits (i.e., grip strength and appendicular lean mass). The relevant summary statistics concerning serum metabolite-associated SNPs with sarcopenia and its indices were calculated from the UK biobank study, using logistic or linear regression models. Age, sex, genotype measurement batch, and 20 genetic principal components were included as covariates in regression analysis to address potential population substructure. UK biobank is a population-based prospective cohort established to allow detailed investigations of the genetic and nongenetic determinants of the diseases of middle and old age.[20] Over 500,000 participants aged 40 to 69 were drawn from 22 assessment centers across the United Kingdom between 2006 and 2010.[23] All participants completed questionnaires on health and lifestyle factors during the baseline survey and provided blood samples for biomarker and genetic assays.” (Pages 3, lines 112, in the clean copy of the revised manuscript) |
||
Comments 3: please include some references and explanations about “Bonferroni correction”. |
||
Response: Many thanks for your comment. We have included more references and explanations about Bonferroni's correction in the manuscript. Action: “To address the issue of multiple testing, multiple-testing-adjusted threshold of the Bonferroni correction was applied to correct for the number of exposures tested for each metabolite and the significance of causal feature was set to P<1.03×10−4 (0.05/486). Serum metabolites were considered as causal features if they reached the Bonferroni adjusted significant level of P<1.03×10−4. Bonferroni’s correction provides a straightforward approach to controlling the type I error rate by dividing the critical level of significance by the number of significance tests performed. However, the correction tends to be conservative if a large number of tests are performed.[29]” (Pages 4, lines 168, in the clean copy of the revised manuscript) |
Reviewer 4 Report
Dear Author,
It is an interesting article. It is extremely useful to identify the factors and pathogenetic mechanisms involved in sarcopenia in order to initiate targeted therapy.
I have a few observations.
In your opinion, what would be the limits of the study ?
Kind regards,
Author Response
Response to Reviewer 4 Comments
|
||
1. Summary |
|
|
Thank you very much for taking the time to review this manuscript. Please find the detailed responses below and the corresponding revisions in the re-submitted files.
|
||
2. Questions for General Evaluation |
Reviewer’s Evaluation |
Response and Revisions |
Does the introduction provide sufficient background and include all relevant references? |
Yes |
Many thanks for your positive evaluation. |
Are all the cited references relevant to the research? |
Yes |
|
Is the research design appropriate? |
Yes |
|
Are the methods adequately described? |
Yes |
|
Are the results clearly presented? |
Yes |
|
Are the conclusions supported by the results? |
Yes |
|
3. Point-by-point response to Comments and Suggestions for Authors |
||
Major comment: It is an interesting article. It is extremely useful to identify the factors and pathogenetic mechanisms involved in sarcopenia in order to initiate targeted therapy. |
||
Response: Many thanks for your positive comments. |
||
Comment 1: In your opinion, what would be the limits of the study? |
||
Response: Thanks very much for your comment. There are several limitations should be recognized in this study. First, it is important to acknowledge that MR approaches may not entirely eliminate residual bias stemming from unmeasured confounders. Second, given the relatively small number of sarcopenia cases in our study, we cannot definitively rule out the possibility that our study may lack sufficient statistical power to detect a subtle association. Third, although this study suggests that some plasma proteins are causally associated with sarcopenia risk. The causal associations and molecular mechanisms still need to be further explored and verified in future studies. We have elucidated all these limitations in the manuscript. Action: “There are several limitations of our study. First, although we have used MR approaches to guard against confounding, we nevertheless cannot fully exclude residual bias due to unmeasured confounders, which is an established limitation of the MR approach.[44] Second, as the number of sarcopenia cases in our study was relatively small, we could not completely rule out the possibility that our study may not have adequate power to discover a weak association. Third, while this study suggests that some metabolites are associated with sarcopenia risk, it primarily provides a prediction without verification. The causal associations and molecular mechanisms still need to be more thoroughly investigated and confirmed in future studies.” (Pages 10, lines 373, in the clean copy of the revised manuscript) |
Round 2
Reviewer 1 Report
Most of the concerns I mentioned in the previous version have been addressed. I have no more concerns about this manuscript.
Moderate English editing would be beneficial.
Author Response
Response to Reviewer 1 Comments
|
||
1. Summary |
|
|
Thank you very much for taking the time to review this manuscript.
|
||
2. Questions for General Evaluation |
Reviewer’s Evaluation |
Response and Revisions |
Does the introduction provide sufficient background and include all relevant references? |
Yes |
Thank you for your evaluation. |
Are all the cited references relevant to the research? |
Yes |
|
Is the research design appropriate? |
Yes |
|
Are the methods adequately described? |
Yes |
|
Are the results clearly presented? |
Yes |
|
Are the conclusions supported by the results? |
Yes |
|
Response to Comments on the Quality of English Language: Moderate English editing would be beneficial. |
||
Response: We appreciate your comment. We have further professionally edited the language of the manuscript. Action: Please see the revised manuscript. |
